# To what extent are the antimalarial markets in African countries ready for a transition to triple artemisinin-based combination therapies?

Freek de Haan[1]*, Oladimeji Akeem Bolarinwa[2], Rosemonde Guissou[3], Fatoumata Tou[4], Paulina Tindana[5], Wouter P. C. Boon[1], Ellen H. M. Moors[1], Phaik Yeong Cheah[6,7], Mehul Dhorda[6,7], Arjen M. Dondorp[6,7], Jean Bosco Ouedraogo[3,4], Olugbenga A. Mokuolu[2], Chanaki Amaratunga[6,7]

1 Copernicus Institute of Sustainable Development, Utrecht University, Utrecht, The Netherlands, 2 College of Health Sciences, University of Ilorin, Ilorin, Nigeria, 3 Institut de Recherche en Sciences de la Sante, Bobo-Dioulasso, Burkina Faso, 4 Institut des Sciences et Techniques, Bobo-Dioulasso, Burkina Faso, 5 School of Public Health, College of Health Sciences, University of Ghana, Accra, Ghana, 6 Mahidol Oxford Tropical Medicine Research Unit, Faculty of Tropical Medicine, Mahidol University, Bangkok, Thailand, 7 Center for Tropical Medicine and Global Health, Nuffield Department of Medicine, University of Oxford, Oxford, United Kingdom

* F.dehaan@uu.nl

**Data Availability Statement:** Data from this study are available upon request. Most interviews that were conducted are directly traceable to individual

## Abstract

### Introduction

Triple artemisinin-based combination therapies (TACTs) are being developed as a response to artemisinin and partner drug resistance in the treatment of falciparum malaria in Southeast Asia. In African countries, where current artemisinin-based combination therapies (ACTs) are still effective, TACTs have the potential to benefit the larger community and future patients by mitigating the risk of drug resistance. This study explores the extent to which the antimalarial drug markets in African countries are ready for a transition to TACTs.

### Methods

A qualitative study was conducted in Nigeria and Burkina Faso and comprised in-depth interviews (n = 68) and focus group discussions (n = 11) with key actor groups in the innovation system of antimalarial therapies.

### Results

Evidence of ACT failure in African countries and explicit support for TACTs by the World Health Organization (WHO) and international funders were perceived important determinants for the market prospects of TACTs in Nigeria and Burkina Faso. At the country level, slow regulatory and implementation procedures were identified as potential barriers towards rapid TACTs deployment. Integrating TACTs in public sector distribution channels was considered relatively straightforward. More challenges were expected for integrating TACTs in private sector distribution channels, which are characterized by patient demand and profit

identities and so we cannot share the interview data without releasing the identities of our respondents. In each interview, respondents introduced themselves and they talked about their direct (working) environment. Moreover, the topics that were discussed and responses that were to our questions, could be directly linked to their job positions and affiliation, especially with higher level policy and regulatory officials. We did guarantee full anonymity to our respondents prior to data collection and so sharing the dataset would be unethical. Upon reasonable request, a list of condensed meaning units or codes can be made available by the MORU Data Access Committee (https://www.tropmedres.ac/units/moru-bangkok/bioethics-engagement/data-sharing).

**Funding:** We are grateful to UK aid and the UK Government's Foreign, Commonwealth & Development Office for financial support. This research was funded in whole, or in part, by the Wellcome Trust [220211]. The funders had no role in study design, data collection, and analysis, decision to publish, or preparation of the manuscript.

**Competing interests:** The authors have declared that no competing interests exist.

motives. Finally, several affordability and acceptability issues were raised for which ACTs were suggested as a benchmark.

## Conclusion

The market prospects of TACTs in Nigeria and Burkina Faso will depend on the demonstration of the added value of TACTs over ACTs, their advocacy by the WHO, the inclusion of TACTs in financial and regulatory arrangements, and their alignment with current distribution and deployment practices. Further clinical, health-economic and feasibility studies are required to inform decision makers about the broader implications of a transition to TACTs in African counties. The recent reporting of artemisinin resistance and ACT failure in Africa might change important determinants of the market readiness for TACTs.

## Introduction

Artemisinin-based combination therapies (ACTs) are the global first-line antimalarial therapies for the treatment of uncomplicated falciparum malaria [1]. A worrying development is the emergence and spread of *Plasmodium falciparum* parasites resistant to both artemisinin and partner drugs, compromising ACT efficacy in large areas of Southeast Asia [2–5]. An even more significant threat is the risk of multidrug resistance spreading further throughout Asia and to the African continent, where most of the malaria burden is clustered [6]. In addition, artemisinin resistance can emerge independently in African countries, as has been increasingly reported in recent years [7]. According to conservative estimates, these scenarios could lead to more than 116.000 excess deaths on an annual base while the economic costs could exceed USD 400 million per year [8].

Innovative solutions are urgently required to restore antimalarial efficacy in areas where ACTs are failing and to protect the rest of the malaria-endemic world from the looming threat of resistance. However, new antimalarial drug compounds are not expected on the market before 2026 [9–11]. A pragmatic short-term solution that could provide long-term benefits is the introduction of triple artemisinin-based combination therapies (TACTS) [12–16]. TACTs combine an artemisinin derivative with two partner drugs, ideally with counteracting resistance mechanisms. This will extend the therapeutic lifetime of the drug combinations, because the parasite will need to develop resistance to three drug compounds instead of two. Previous studies have shown promising results [17] and several TACT combinations are now being developed and tested to translate this potential into end-products [15]. Once they are confirmed to be safe, tolerable, efficacious and non-inferior to ACTs, these TACTs could provide direct clinical relief in Southeast Asia. Moreover, a rapid and sustainable transition to TACTs in Africa could mitigate the risk of spread and *de novo* emergence of artemisinin and partner drug resistance on the continent.

Despite these promising developments, history has shown that changing first-line malaria therapies is a time and resource intensive process. Previous drug transitions have been slow and challenging, even when new therapies were clinically superior to failing alternatives [18–21]. Challenges have, amongst others, been associated with supply-side and demand-side factors and with underperforming healthcare systems in endemic countries [22].

A complicating factor for a transition to TACTs in Sub-Saharan Africa is that current ACTs are still effective for the treatment of falciparum malaria. TACTs would not provide direct

additional clinical benefit to individual patients, but rather benefit the larger community and future patients by mitigating the risk of the emergence and spread of drug resistance. Engaging in TACTs thus requires direct investments at the country level, whilst the benefits of TACT deployment would be long-term and transcend national borders. This, in combination with lessons learned from the problematic introduction of previous therapies, warrants an assessment of the readiness of the antimalarial drug markets in African countries for a transition to TACTs [15, 23]. Similar anticipatory studies have not been conducted for innovative antimalarial therapies and findings could support strategic decision making in the battle against drug-resistant malaria. This study aims to explore the extent to which stakeholders perceive the antimalarial drug markets in African countries ready for a transition to TACTs.

## Materials and methods

### Research design

The study was conducted under the auspices of the UK Government's Foreign, Commonwealth & Development Office funded Development of Triple Artemisinin Combination Therapies (DeTACT) project and explores the readiness of antimalarial drug markets in African countries for a transition to TACTs [15]. An initial pilot study was conducted in Nigeria and Burkina Faso, two countries that suffer from high malaria endemicity [6], yet have different healthcare systems. Table 1 provides demographic and malaria-related data for Nigeria and Burkina Faso. We used a multiple case design, that allows investigating the country-specific contexts, while enabling the extraction of overarching themes [24]. The study employs a qualitative approach of data collection, involving in-depth interviews and focus group discussions (FGDs) with key stakeholders.

### Theoretical and thematic approach

The study uses an 'innovation systems' approach to explore the readiness of African countries for a transition to TACTs [25]. The innovation systems approach assumes that integrating new technology in society is a collective effort that takes place in a complex social system. To explain or predict the success of an innovative therapy such as TACTs, one should not only consider medical-technological characteristics, but also the dynamics of the surrounding innovation system [22, 26]. An innovation system consists of all actors, networks and institutions involved in the development, distribution and utilization of the therapy.

We identified the key actor groups in the innovation system of antimalarial therapies (Fig 1). Our study focuses on country-level dynamics, and therefore the national borders are demarcated by a dashed line. To prepare for data collection, we conducted a thematic analysis in which we identified the major barriers and enablers of previous antimalarial drug transitions from literature review [27]. These barriers and enablers were grouped and assigned to the key actors in the innovation system. Finally, we constructed semi-structured interview guides per key actor group to facilitate data collection [28].

### Respondent selection

Respondents at different levels of the innovation system of antimalarial therapies (Fig 1) in Nigeria and Burkina Faso were identified and invited to participate in interviews and FGDs. We divided the key actor groups into three overarching categories to facilitate data analysis: Policy makers, Suppliers, and End-users. The rationale for this categorization was that they combine an overview of the innovation system with knowledge on specific parts of the system. Policy makers are best positioned to reflect on policy procedures and the regulatory trajectory.

**Table 1. Demographic and malaria data on the settings in Nigeria and Burkina Faso.**

| | | Nigeria | Burkina Faso |
|---|---|---|---|
| **Demographic** | Population (2019) [a] | 200.9 M | 20.2 M |
| | First language | English | French |
| | Capital | Abuja | Ouagadougou |
| | Human Development Index (2019) [a] | #158 | #182 |
| | GNI (PPP) (2019) [a] | USD 5.170 | USD 2.220 |
| | Population > USD 1.90 PPP (2019) [a] | 53.5% | 43.7% |
| **Health -demographic** | Life expectancy (2018) [a] | 54 YRS | 61 YRS |
| | <5 mortality per 1000 births (2019) [a] | 117.2 | 87.5 |
| | Health spending per capita (2018) [a] | USD 83.75 | USD 40.25 |
| | Health spending % GDP (2018) [a] | 3.89% | 5.63% |
| **Malaria** | Drug regulation authority | National Agency for Food and Drug Administration and Control (NAFDAC) | Agence Nationale de Régulation Pharmaceutique (ANRP) |
| | Malaria control program | National Malaria Elimination Program (NMEP) | Programme National de Lutte contre le Paludisme (PNLP) |
| | Population at risk of malaria (2018) [b] | 100% | 100% |
| | Estimated malaria cases (2018) [b] | 57.2 M | 7.9 M |
| | Estimated malaria deaths (2018) [b] | 95.844 | 12.725 |
| | First line anti-malarial therapy | Artemether-lumefantrine | Artemether-lumefantrine |
| | | Artesunate-amodiaquine | Artesunate-amodiaquine |
| | | | Dihydroartemisinin-piperaquine |
| | ACT in guidelines since | 2004 | 2005 |
| | ACT for free in public sector? | Yes | Only for children <5 years and pregnant women |
| | Antimalarials prescribed over-the-counter? | Yes | Yes |

[a] https://data.worldbank.org/.

[b] WHO malaria report 2019.

Suppliers are most knowledgeable about distribution, retail and prescription issues, while End-users have the most relevant insights on the actual utilization of new therapies. Sampling of respondents in each country continued until data saturation was reached. The full list of respondents is provided in Table 2.

## Data collection

Preparatory meetings were held among members of the research team prior to data collection to discuss the study aims and to prepare material for data collection. Pilot interviews were conducted to further improve mutual understanding and to reduce ambiguity. Using the semi-structured interview guides enabled flexibility to new topics while still covering the same topics between the settings and respondent groups.

Each interview and FGD was conducted by at least two interviewers; one asking questions and the other taking notes. Interviews and FGDs were conducted in prearranged venues where there were no distractions and respondents were able to express themselves freely. On average 8 people participated per FGD. Each interview and FGD lasted between 30 and 120 minutes and were recorded with the consent of the respondent(s).

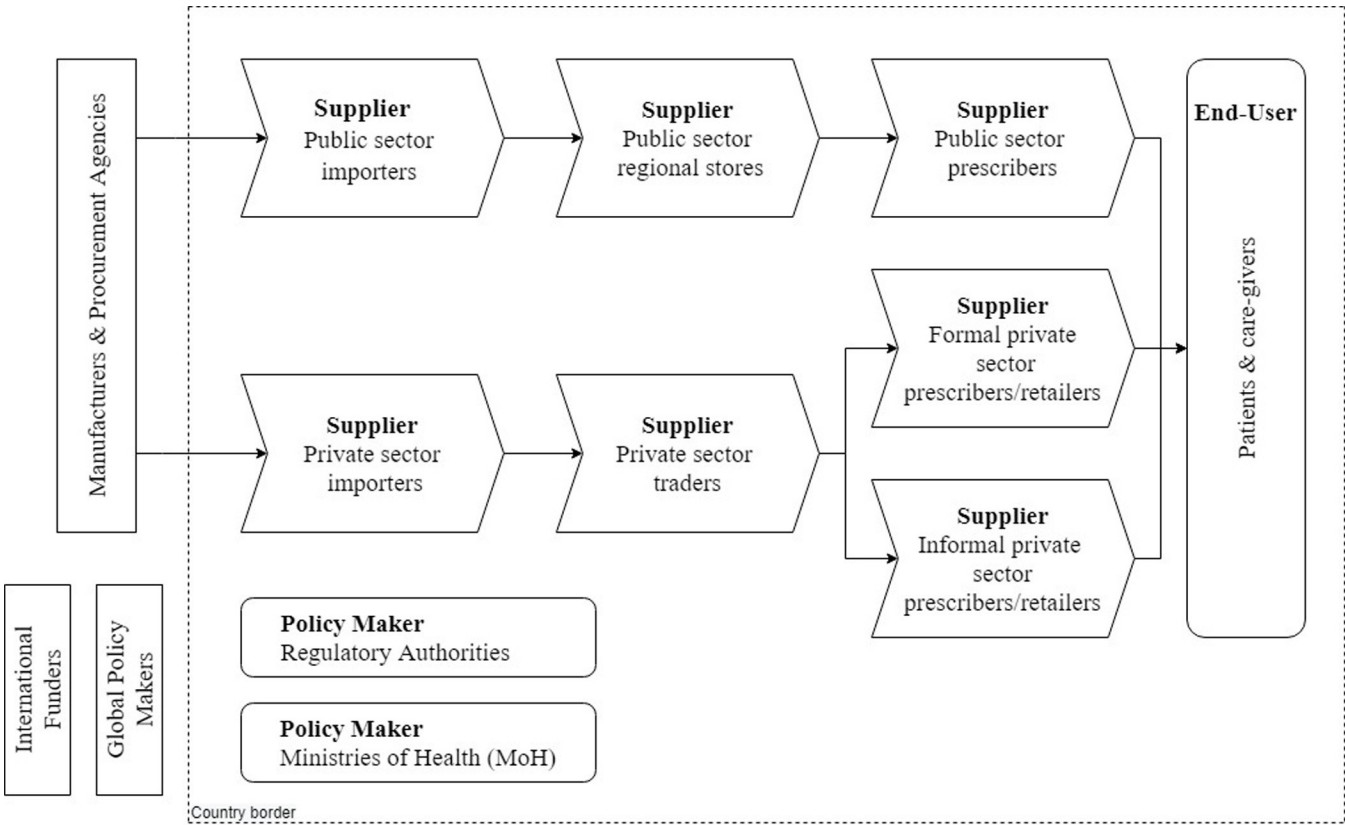

**Fig 1. Key actor groups in the innovation system of antimalarial therapies.**

## Data analysis

The tape recordings were transcribed and all non-English transcripts were translated to English. The transcripts were uploaded to NVivo12 Pro software and subjected to a process of coding. The coding process comprised both deductive and inductive techniques. A codebook was constructed based on the prior thematic analysis (deductive). New themes were developed for data that did not match the existing codes but were considered relevant for the study (inductive). The interview transcripts were coded independently by two researchers who analyzed all transcripts and discussed emerging themes. In a second round of coding, all themes were merged into overarching categories, the reflections were extracted per respondent group, and storylines were written.

## Ethical approval

In Nigeria, ethical approval from the Institutional Review Board was obtained from the University of Ilorin Teaching Hospital (approval number ERC/PAN/2019/07/1916). In Burkina Faso, ethical approval was obtained from the Institutional Ethics Committee for Health Research (CEIRES). The Oxford Tropical Research Ethics Committee (OxTREC) approved the overall research project (approval number 552–19). Written participant consent was obtained prior to each interview and FGD. Respondents were informed about the objectives of the study and they were asked to sign a consent form. Permission to mention the affiliation of the respondent and to audio record the conversation was asked verbally.

**Table 2. Respondents in Nigeria and Burkina Faso.**

| Country | Category | Number | Background / affiliation | Interview/FGD |
|---|---|---|---|---|
| **Nigeria** | **Policy Makers** | 1 | Principal malaria researcher | Interview |
| | | 3 | Regulatory officials | Interview |
| | | 6 | National malaria policy officials | Interview |
| | | 1 | Regional malaria policy official | Interview |
| | **Suppliers** | 4 | Public sector prescriber | Interview |
| | | 4 | Private sector prescriber | Interview |
| | | 3 | Village Health Workers | FGD |
| | | 2 | Public sector wholesaler /retailers | Interview |
| | | 5 | Private sector wholesaler/ retailers | Interview |
| | **End-users** | 7 | Community members | Interview |
| | | 4 | Community members | FGD |
| **Burkina Faso** | **Policy Makers** | 2 | Principle malaria researcher | Interview |
| | | 2 | Regulatory officials | Interview |
| | | 3 | National malaria policy officials | Interview |
| | | 3 | Regional malaria policy officials | Interview |
| | **Suppliers** | 5 | Public sector prescriber | Interview |
| | | 9 | Private sector prescriber | Interview |
| | | 3 | Public sector wholesaler /retailer | Interview |
| | | 8 | Private sector wholesaler/ retailer | Interview |
| | **End-users** | 4 | Community members | FGD |

## Results

This study explored market readiness for a transition to TACTs by probing stakeholders' perceptions in two African countries–Nigeria (NG) and Burkina Faso (BF). The findings from interviews and FGDs are categorized by the eight themes that emerged during data analysis.

### Market approval, regulation and domestic production

Policy makers in Nigeria and Burkina Faso did not foresee major challenges in the regulatory trajectory for TACTs. Obtaining market approval for TACTs would be relatively straightforward under the assumption of supportive safety and efficacy data from clinical trials. It was, however, acknowledged that the trajectory towards market approval can be a lengthy one, while in the context of rapidly increasing drug resistance shorter timelines may be needed. [*"I think generally we are trying to streamline the process of registering medicines in Nigeria, we are not where we want to be but it is a lot shorter than it used to be"*—Policy maker 7, regulatory official, NG]. Internal capacity issues at the regulatory authority and submission of incomplete dossiers were cited as common causes of delays in the regulatory trajectory. Early submission of dossiers was suggested to be the most feasible way for obtaining rapid market approval for TACTs.

Respondents mentioned several challenges they encounter in the performance of the antimalarial drug market in their countries. It was acknowledged that antimalarials are often still being prescribed over-the-counter and beyond the official channels. As a result, substandard and outdated medicines such as chloroquine are still commonly deployed in both countries. Several policy makers and suppliers mentioned that market surveillance systems remain weak and that this could be a barrier to the uptake of TACTs. [*"A threat can be that there are too many medicines on the market and lack of supervision. Then there may be no incentive to choose*

*for TACT instead of another option"*–Supplier 4, public sector clinician, NG]. The limited market surveillance and the low levels of regulatory compliance imply that patients can purchase the medicines they favor rather than those that are recommended in treatment guidelines. Challenges were particularly associated with the (informal) private sectors and respondents admitted that introducing TACTs under these circumstances is difficult. General health system reforms and increased investments in market regulation were cited as potential solutions.

Policy makers and suppliers in Nigeria emphasized that the domestic pharmaceutical industry is an important source of antimalarial drugs. Engaging domestic manufacturers in producing TACTs was therefore considered crucial to stimulate its uptake in Nigeria. A repeatedly cited challenge in this regard is that Nigerian manufacturers have not yet been able to produce according to global quality standards, including Good Manufacturing Practice standards. [*"The local manufacturers produce all the ACTs, but the barrier is that none of them is WHO pre-qualified"*–Policy maker 2, national malaria policy official, NG]. This implies that domestic manufacturers cannot become eligible to international donor subsidies. Respondents in Burkina Faso indicated that there is no domestic industry for the production of antimalarial therapies. Adopting TACTs as first-line therapy would therefore require adaptations in import and procurement procedures. Some suppliers perceived a potential transition to TACTs as an opportunity to start domestic production of antimalarials in Burkina Faso. [*"If we have local production, I'm not saying that there won't be a shortage, but there is a certain guarantee. We will always be able to put pressure for more drugs."*–Supplier 13, private sector clinician, BF].

## Inclusion in treatment guidelines and subsidy arrangements

All respondent groups agreed that including TACTs in national treatment guidelines would be critical for its future uptake in Nigeria and Burkina Faso. Several policy makers emphasized that WHO endorsement would be a major enabler for inclusion in these national treatment guidelines. [*"WHO is a benchmark in management, it is a guide, it is a reference, all documents and national guidelines take reference from the WHO"*–Supplier 22, private sector clinician, BF]. Most policy makers and suppliers were skeptical about a transition towards TACTs as long as ACTs are not failing within their borders. [*"I'm not too much in favor of changing as long as ACTs are still effective. So we should continue to use them until they become resistant. If it doesn't work, then we can change!"*- Supplier 21, private sector clinician, BF].

Changing policy now to avoid resistance in the future was considered an insufficient argument to enact. This position was, however, contested by some other respondents who preferred pro-active anticipation by the government instead of passively waiting until artemisinin and partner drug resistance has been detected. [*"We should not be waiting for the emergence of resistance to happen, before steps are taken"*–Supplier 9, village health worker, NG]. Some respondents argued that the introduction of TACTs should be guided by actual data that confirms the presence of antimalarial drug resistance. *["So, countries like Nigeria want to document local evidence. You know, they want local evidence of the resistance before they could accept it."*– Policy maker 4, national malaria policy official, NG]. A transition to TACTs would then require a strategic plan which would contain specific policy actions at pre-defined resistance thresholds.

Policy makers, suppliers and end-users emphasized the importance of subsidy arrangements for deploying TACTs. [*"You just have to integrate it into your program and subsidize it and make it available."*–Supplier 12, public sector clinician, BF]. Since price dictates consumer preferences, they expected TACTs to be unaffordable for large proportions of the population without external financial support. This was especially considered relevant for private sector deployment because of the limited availability of subsidy arrangements. For public sector

deployment, willingness of the Global Fund to Fight Aids, Tuberculosis and Malaria (GFATM) to purchase TACTs was considered critical by policy makers and suppliers. In Nigeria, inclusion of TACTs in national insurance arrangements such as the National Health Insurance Scheme (NHIS) was suggested as another tool to promote uptake.

## Implementation programs

Implementation refers to the process of translating policy decisions into practice. Respondents in Nigeria and Burkina Faso agreed that deliberate implementation programs would be required for TACTs if it becomes the first-line therapy for the treatment of malaria. The National Malaria Elimination Program (NMEP) in Nigeria and the Programme National de Lutte contre le Paludisme (PNPL) in Burkina Faso would be responsible for coordinating a prospective implementation program. Respondents acknowledged that the implementation of ACTs had been slow and that outdated medicines persistently remain available on the market. According to them, lessons learned should be used to inform prospective implementation strategies for TACTs.

Implementation programs should first of all inform retailers (e.g. pharmacists, shop owners, drug sellers) and prescribers (e.g. clinicians, nurses) about the implications of switching to TACTs. [*"The professionals must be involved: the nurses, doctors, pharmacists."*–Policy maker 6, regulatory official, BF]. Not only do these retailers and prescribers deliver the drugs to patients, they are also responsible for information dissemination to patients. [*"Who deploy the ACTs? The healthcare providers. So, you don't neglect them."*–Policy maker 2, national malaria policy official, NG]. Implementation programs should be transparent about the added benefits of TACTs in terms of delaying drug resistance and they should cover an extended period of time. [*"The strategies to be used is what I've just said: advocacy, communication, mobilization, sensitization and you don't limit it to just saying you've done it [. . .]. It has to be continuous for a certain period of time."*–Policy maker 5, national malaria policy official, NG].

Another component of an implementation strategy would be information campaigns to inform the general population. Several tools to convey the message were proposed, including radio, television, billboards, and social media channels. Posters in pharmacies were suggested as another, more targeted, way to notify malaria patients. [*"The population must not be left behind, they are the target, they must be informed. They must be informed through the different channels: radio, TV, billboards."*–Supplier 20, private sector wholesaler, BF]. Involving religious organizations and community leaders was suggested to convey the message to hard-to-reach populations. Moreover, it was deemed important to adapt information campaigns in order to reach illiterate populations. Implementation strategies should aim for early stage sensitizing and informing the population about the appropriate use of TACTs.

## Public sector distribution

Policy makers and suppliers in Nigeria and Burkina Faso did not predict significant challenges for integrating TACTs in public sector distribution chains. Adoption of TACTs by global and national policy-, and donor organizations was considered a decisive factor to the public sector uptake of TACTs. [*"The commodities in the public sector is mainly determined by the support from donors. And the donors procure this drug and supply it to the country. So the focus will need to be paid towards the donor."*–Policy maker 2, national malaria policy official, NG]. Respondents acknowledged that a transition to TACTs would require adaptations in importing, procurement and distribution practices, which would be relatively straightforward because public sector forecasting and stock-taking procedures for antimalarial therapies have been well-established.

Respondents expressed that there are no financial incentives for public-sector actors to deviate from guidelines, in contrast to the private sector. Some respondents mentioned that introducing TACTs in the public sector may even be an effective strategy to promote TACTs uptake in the private sectors. [*"The acceptance in the public setting is what dictates the acceptance in the private."*–Supplier 14, public sector wholesaler, NG]. In Burkina Faso, several suppliers emphasized the role of the Centre d'Achat des Médicaments Essentiels Génériques (CAMEG) for public sector introduction of TACTs. The CAMEG is a public organization that dictates the public sector procurement and distribution of medicines in Burkina Faso. The CAMEG exclusively focuses on the deployment of generic therapies and therefore it was suggested that TACTs should be introduced under the conditions of generic drugs rather than as a branded proprietary drug. [*"The antimalarial drugs in the public sector are essentially generics, unlike the private sector where we have the specialized drugs."*–Supplier 12, public sector clinician, BF].

## Private sector distribution

In comparison to the public sector equivalent, more challenges were anticipated for integrating TACTs in private sector distribution channels in Nigeria and Burkina Faso. Private sector stocking decisions are usually based on patient demand and profit motives rather than treatment guidelines. [*"Firms go for customer satisfaction. They probe to see how the patient can be satisfied."*–Policy maker 6, regulatory official, BF]. Policy makers and suppliers stressed that, once introduced, TACTs will have to compete with alternative therapies in the private sector markets. They were skeptical whether mitigating the risk of resistance would be a sufficient reason for the private sector to engage in TACTs as long as further market regulation remains absent. [*"Why would a profit maker go and procure what is unlikely to sell*?"–Policy maker 2, national malaria policy official, NG]. Enhancing market regulation practices, financial rewards for prescribers, and artificial stimulation of demand were proposed as strategies to engage private sector actors into a transition to TACTs. An example of artificial demand stimulation would be subsidies in the early stages of deploying TACTs.

Policy makers and suppliers considered an instant transition to TACTs in the private sector unrealistic in terms of adapting importation, procurement and supply procedures. [*"We just need some time to sell off the old stock."*–Supplier 1, private sector wholesaler, BF]. Many wholesalers and retailers in Nigeria and Burkina Faso have invested in contracts with providers higher up in the supply chain. They advocated for a transition period in which TACTs are gradually implemented while ACTs are simultaneously phased out. This was considered a more viable strategy because it would allow them to clear their existing stocks and pending orders. Since most antimalarial drugs have a shelf-life of three years, a three-year period was proposed as an appropriate transition period by one supplier. Other suppliers disagreed with the necessity of a transition period stating that people will not be interested in TACTs as long as ACTs are still available on the market and therefore they preferred a direct removal of ACTs. [*"If the other one stays, there is a chance that there will be resistance. In order to protect the TACTs, the ACTs must first be removed."*–Supplier 13, private sector clinician, BF].

## Retail and prescription

Several respondents anticipated challenges in engaging retailers and prescribers as long as TACTs are not clinically superior to ACTs and as long as alternative antimalarials remain available on the market. Some suppliers expressed that retailers and prescribers had been overlooked in the transition from monotherapies to ACT and this had contributed to limited compliance at the time. The continuing availability of chloroquine in both countries was given as an example of the ongoing challenges.

Several policy makers and suppliers mentioned that retailers and prescribers are not always aware of the risks and implications of drug resistance, which could affect negatively their preparedness to switch to TACTs. Hence, training programs were considered essential in the early stages of deploying TACTs. These training programs should be clear and transparent about the risks of resistance and the potential benefits of TACTs. [*"Prescribers must be trained and informed about the added value and the reasons for the change."*–Supplier 6, public sector wholesaler, BF]. Conferences and educational programs were also considered suitable platforms to inform and engage prescribers and retailers. Positive experiences from other settings such as Southeast Asia could be a boost to the perceived credibility of TACTs. It was also suggested that prescribers and retailers should be given elaborate information about dosing and side-effects of TACTs, because they are responsible for further dissemination of such information to patients and caregivers. Some suppliers and end-users were concerned that private sector retailers will adjust their prices once demand for TACTs would increase.

## Affordability

Cost-issues were considered important for a successful transition to TACTS by all respondent groups in Nigeria and Burkina Faso. [*"If you come with prices like 15000 f a box, people will leave you and your drug."*—Policy maker 1, principal malaria researcher, BF]. Several policy makers and suppliers argued that TACTs should be introduced for subsidized prices because production prices would probably be too high for many patients. Subsidies were also considered a requirement to effectively compete with ACTs and chloroquine. One policy maker stated that, once affordability for TACTs is ensured, all other acceptance issues will follow. Others argued that price is not the most important determinant for effective medication of a life-threatening disease. [*"If the drug is really effective according to them, if it is really effective, whatever the price, they pay to be liberated. Health is priceless."*–Policy maker 8, regional malaria policy official, BF].

Respondents agreed that prices of TACTs should not exceed those of present ACTs, indicating patients will tend to go for the cheapest option. [*"If the price is slightly ahead of the current ACT, sir, it is dead on arrival."*–Policy maker 10, regional malaria policy official, NG]. One supplier contrasted this view, claiming that low prices of TACTs may be perceived as an indicator of limited quality and may therefore foster distrust. End-users did not consider costs as the main determinant for their willingness to adopt TACTs, and are prepared to pay more if the drug has a reputation of high efficacy and limited side-effects. [*"Money is not the most important thing because what we are really interested in is the well-being of our children and ourselves."*–End-user FGD 2, community member, NG].

## Acceptance

In addition to clinical efficacy and affordability, all respondent groups highlighted that side-effects can be a potential barrier to TACTs uptake. In particular nausea and vomiting were mentioned as undesirable side-effects that could negatively affect acceptability. These side-effects are especially associated with amodiaquine and therefore some suppliers voiced their preferences for TACTs without this drug compound. [*"I take an example: artesunate-amodiaquine, these are combination therapies that were used at one time but very quickly it was realized that people were not tolerating the side effects and then we switched to artemether-lumefantrine."*–Supplier 7, public sector wholesaler BF]. Other suppliers and most end-users claimed that side effects are an integral aspect of antimalarial therapies. Mild nausea would not prevent end-users from engaging in TACTs and they argued that medication exists for mitigating most side-effects. Respondents emphasized that side-effects are acceptable as long as TACTs

are effective and affordable. ["*If I am going to be taking that, it is my own best interest. When it works fast then there is no problem about that at all.*"–End-user FGD 1, community member, NG].

Other acceptance issues raised were: tablet size and shape, taste, and the number of pills. For all these acceptance issues, the current ACT treatment regime was given as a benchmark: new drugs should not be more expensive, should not cause more side-effects, should not contain more pills and should not have a more bitter taste than ACTs. ["*As far as taste is concerned, it shouldn't be too bitter and the size should be the same as what it is present.*"–End-user FGD 2, community member BF]. Finally, respondents suggested that branding and attractive packaging could positively affect community attitudes towards TACTs.

## Discussion and conclusions

The goal of the study was to explore market readiness for a transition to TACTs in African countries by exploring stakeholders' perceptions. A qualitative study was conducted in Nigeria and Burkina Faso and comprised in-depth interviews and FGDs with key actor groups at different levels of the antimalarial innovation system. A number of barriers and enablers towards deploying TACTs emerged from the data. The study revealed that the market prospects of TACTs in Nigeria and Burkina Faso will depend on the demonstration of the added value of TACT over ACTs. National decision makers are unlikely to initiate a transition to TACTs in order to mitigate the risk of drug resistance. Instead, they will await endorsement by global institutes, in particular the World Health Organization (WHO), and funding decisions by international donors such as the Global Fund to fight Aids, Tuberculosis and Malaria (GFATM). This implies that these global-level institutes have a major responsibility in navigating the global fight against drug-resistant malaria. They should consider proactive changes to treatment policies to prevent similar delays in global coordination as those encountered during the transition to ACTs in the early 2000s [29]. This is all the more relevant now that genetic markers of artemisinin resistance have been observed in Rwanda, Uganda and Tanzania [20, 30–35] while inadequate efficacy of artemether-lumefantrine, the most commonly used antimalarial in Africa and elsewhere, was recently reported in Angola and Burkina Faso [36–38]. These findings imply an urgent need to identify and develop alternative treatment options.

At the national levels, several barriers were raised that would affect the market prospects for TACTs in Nigeria and Burkina Faso. Respondents referred to lengthy regulatory and implementation procedures [18, 19, 39] and the ongoing challenges with regards to the private sector markets for antimalarial therapies [40–42]. In particular the persistent demand for chloroquine and over the counter prescription of antimalarial therapies were repeatedly mentioned. The limited (private sector) compliance to first-line therapies and the widespread availability of substandard, outdated or even counterfeit drugs has been reported in several African settings by the ACTwatch group [40, 43–45]. In this context of suboptimal treatment practices, it is warranted to question whether scarce resources should be dedicated to a transition to TACTs [15, 16] or rather be invested in improving current malaria management practices [46, 47]. From a health-economic perspective, investing in mitigating the risk of resistance seems justified, as the costs of a drug transition is unlikely to exceed the costs that would be provoked by artemisinin and partner drug resistance [8, 20, 48]. Further mathematical modeling and feasibility studies are required to determine if TACTs in Africa are indeed the optimal way forward to address the threat of artemisinin and partner drug resistance.

Relatively few challenges towards implementing TACTs in public sector distribution channels were identified, under the assumption of support by policy makers and donor funders. This aligns with other studies that have demonstrated that public distribution channels are

relatively adaptive for new generations of antimalarial therapies [43, 49]. More challenges were expected for engaging private sector stakeholders in a transition to TACTs. In these private sector channels, TACTs will have to compete with alternative treatments on the market and switching to TACTs can be perceived as misaligned with profit motives of distributors, retailers and prescribers. Similar private sector challenges have previously been reported for innovative therapies for malaria [50] and other poverty-related diseases [51–53] under the pressure of drug resistance. Proposed solutions from the literature include intensified market regulation [49, 52] and increased investments in private sector subsidy arrangements [41, 42, 48].

Several affordability and acceptance issues for TACTs were raised, for which current ACTs were suggested as a benchmark. Respondents advised that TACTs should not be more expensive, cause more side-effects, contain more pills and have a more bitter taste than ACTs. Furthermore, information campaigns should be transparent on the motives and the rationale of a prospective switch to TACTs. In previous studies, adoption decisions by malaria patients have been associated mostly with the availability of a particular therapy [54, 55]. Our study adds to the limited knowledge on factors that affect the actual acceptance of innovative antimalarial therapies [56].

The innovation systems approach was applied to study the feasibility of a transition to TACTs in Nigeria and Burkina Faso. This approach has become a well-established analytical tool in transition studies for generating insight into innovation barriers and defining policy implications accordingly [57]. Similar systemic perspectives have increasingly been applied in addressing (global) health challenges and have been valued for their integral perspective and their explanatory power [26, 58–60]. The innovation systems approach has particularly been applied for understanding sustainability transitions where complex trade-offs need to be made between short-term investment and long-term benefits. This is exactly the case for TACTs in Africa, where investments (national governments, individual patients) and benefits (entire malaria world, future patients) are not well-aligned [23]. The systemic approach demonstrated a complex set of links, relations and interdependencies between actors, networks and institutions. For example, engagement by global and national decision makers is required for a transition to TACTs but it is not sufficient if prescribers and patients reject the use of TACTs. Similarly, positive attitudes from end-users will not be relevant unless well-functioning distribution channels are established.

The innovation systems approach indicates that a transition to TACTs would demand collective actions at all levels of the anti-malarial innovation system (e.g. at the level of development, diffusion and actual deployment) [22]. Moreover, the introduction of TACTs would require alignment with the financial, regulatory and other institutional frameworks that are prevalent for innovative antimalarial therapies. Addressing these multi-faceted issues is complex and requires strong policy coordination and systemic tools and roadmaps [61–64].

## Way forward

We presented a pilot study to assess the feasibility of a transition to TACTs in two African countries. Several barriers and enablers towards deploying TACTs were found in Nigeria and Burkina Faso, which can be used to inform strategic decision-making in the battle against drug-resistant malaria. Although some of these insights may be applicable to other settings, the generalizability of this study is limited. African countries are heterogeneous in nature and they are characterized by different healthcare systems. Therefore, similar bottom-up studies on the feasibility of deploying TACTs are required in other African settings. Such studies should be prioritized in countries reporting artemisinin resistance and/or ACT failure and countries with increased risk of resistance due to their geographic location or epidemiological

situation. Moreover, the situation of drug resistance is evolving and ACT failure was not yet confirmed in Africa by the time of data collection. The recent increased reporting of artemisinin resistance and ACT failure in Africa might change important determinants of the market readiness for TACTs.

Although this study focused on country-level dynamics, the actual prospects of deploying TACTs are to a large extent determined by global decision makers such as the WHO, international funders like GFATM and support from the Medicines for Malaria Venture (MMV) in coordinating drug development. In addition, the pharmaceutical industry plays a central role, and will be required to translate the concept of TACTs into end-products and to ensure its production in sufficient quantities. Encouraging are the successes that have been achieved in the past: a supportive global landscape for the development and deployment of new antimalarial therapies is now well-established [22]. We can draw from these experiences in terms of product-development partnerships, funding mechanisms, regulatory arrangements and intellectual property management. This will, however, require early sensitizing and engagement with global-level policy makers, pharmaceutical industry, and other stakeholders.

Finally, although the study provides insight into factors that are likely to affect the introduction of TACTs in two African countries, it does not provide final answers on whether TACTs should be introduced in Africa. A further rapid increase in artemisinin resistance in African countries would change many of the scenarios described in this study, and would render TACTs one of the last remaining treatment options for multidrug-resistant falciparum malaria. Also, further clinical, health-economic and feasibility studies are required to build an evidence base to inform decision makers on the implications of deploying TACTs to delay or prevent artemisinin resistance and/or ACT failure. Such studies are now underway within the Development of Triple Artemisinin Combination Therapies (DeTACT) project [15] which combines clinical, mathematical modeling, ethical and market positioning research.

## Supporting information

**S1 File. Interview guides in English and French.**
(ZIP)

## Acknowledgments

We thank the interviewees for their valuable input.

## Author Contributions

**Conceptualization:** Freek de Haan, Paulina Tindana, Phaik Yeong Cheah, Chanaki Amaratunga.

**Data curation:** Freek de Haan.

**Formal analysis:** Freek de Haan.

**Funding acquisition:** Arjen M. Dondorp.

**Investigation:** Freek de Haan, Oladimeji Akeem Bolarinwa, Rosemonde Guissou, Fatoumata Tou, Paulina Tindana, Phaik Yeong Cheah, Mehul Dhorda, Jean Bosco Ouedraogo, Olugbenga A. Mokuolu, Chanaki Amaratunga.

**Methodology:** Freek de Haan, Paulina Tindana.

**Project administration:** Freek de Haan, Wouter P. C. Boon, Ellen H. M. Moors, Mehul Dhorda, Arjen M. Dondorp, Jean Bosco Ouedraogo, Olugbenga A. Mokuolu, Chanaki Amaratunga.

**Resources:** Freek de Haan, Oladimeji Akeem Bolarinwa, Rosemonde Guissou, Fatoumata Tou, Paulina Tindana, Wouter P. C. Boon, Ellen H. M. Moors, Phaik Yeong Cheah, Mehul Dhorda, Arjen M. Dondorp, Jean Bosco Ouedraogo, Olugbenga A. Mokuolu, Chanaki Amaratunga.

**Supervision:** Wouter P. C. Boon, Ellen H. M. Moors, Mehul Dhorda, Arjen M. Dondorp, Jean Bosco Ouedraogo, Olugbenga A. Mokuolu, Chanaki Amaratunga.

**Validation:** Freek de Haan, Paulina Tindana.

**Visualization:** Freek de Haan.

**Writing – original draft:** Freek de Haan.

**Writing – review & editing:** Freek de Haan, Oladimeji Akeem Bolarinwa, Rosemonde Guissou, Fatoumata Tou, Paulina Tindana, Wouter P. C. Boon, Ellen H. M. Moors, Phaik Yeong Cheah, Mehul Dhorda, Arjen M. Dondorp, Jean Bosco Ouedraogo, Olugbenga A. Mokuolu, Chanaki Amaratunga.

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
