## [Decision Letter · Decision Letter 0]

28 Jun 2021

PONE-D-21-13692

To what extent are the antimalarial markets in African countries ready for a transition to Triple Artemisinin-based Combination Therapies?

PLOS ONE

Dear Dr. de Haan,

Thank you for submitting your manuscript to PLOS ONE. After minor revisions, we would like to accept this manuscript. We invite you to submit a revised version of the manuscript that addresses the points raised during the review process.

We look forward to receiving your revised manuscript.

Kind regards,

Lucy C. Okell

Academic Editor

PLOS ONE

Journal Requirements:

3. Thank you for including your ethics statement:  "In Nigeria, ethical approval from the Institutional Review Board was obtained from the University of Ilorin Teaching Hospital (approval number ERC/PAN/2019/07/1916). In Burkina Faso, ethical approval was obtained from the Institutional Ethics Committee for Health Research (CEIRES). The Oxford Tropical Research Ethics Committee (OxTREC) approved the overall research project (approval number 552-19).".   

Please provide additional details regarding participant consent. In the ethics statement in the Methods and online submission information, please ensure that you have specified what type you obtained (for instance, written or verbal, and if verbal, how it was documented and witnessed). If your study included minors, state whether you obtained consent from parents or guardians. If the need for consent was waived by the ethics committee, please include this information.

4. Please include a copy of the interview guide used in the study, in both the original language and English, as Supporting Information, or include a citation if it has been published previously.

Reviewers' comments:

Reviewer's Responses to Questions

**Comments to the Author**

1. Is the manuscript technically sound, and do the data support the conclusions?

Reviewer #1: Yes

Reviewer #2: Yes

2. Has the statistical analysis been performed appropriately and rigorously? 

Reviewer #1: N/A

Reviewer #2: Yes

3. Have the authors made all data underlying the findings in their manuscript fully available?

Reviewer #1: Yes

Reviewer #2: No

4. Is the manuscript presented in an intelligible fashion and written in standard English?

Reviewer #1: Yes

Reviewer #2: Yes

5. Review Comments to the Author

Reviewer #1: The topic is timely and vital. Plasmodium resistance to antimalarial medicines is one of the key persistent challenges in the fight against malaria. Although there is no convincing evidence of plasmodium resistance to ACTs in Nigeria and Burkina Faso, it is imperative to plan and seek ways to mitigate this challenge in the future.

The study was guided by the innovative systems approach and considered multiple stakeholders.

Respondent selection

How were participants selected? Was this a purposive or convenience sampling?

Any reasons why some people from pharmaceutical/drug manufacturing industry were not interviewed?

Data collection

Although you mentioned your semi-structured interview guide enabled flexibility, one would imagine that the questions the authors asked policymakers would be hugely different from the questions for end-users. How did you deal with this? For example, did you have separate questions for each respondent group? Did you have the same interview guides for focus groups and interviews?

Data analysis

How many people coded the data? Did you evaluate the Inter-rater reliability?

Results

The participant quotations presented illustrate the themes/findings; however the supplier category is a very diverse group . For example, the village worker may perhaps have a different opinion from a public sector prescriber. It will be helpful for the reader to have the IDs reflect this. So instead of having Supplier #22, you can have public sector prescriber #... or Village worker #....

Reviewer #2: The authors have explored the extent to which the antimalarial drug markets in African countries are ready for a transition to TACTs piloting with Nigeria and Burkina Faso. This is posited in the African setting where ACTs are still effective and TACTs would not additionally benefit the individual patient but rather future patients and the larger community by delaying or preventing artemisinin and partner drug resistance. They show that market prospects of TACTs in Nigeria and Burkina Faso will depend on the demonstration of the added value over current ACTs, their inclusion in financial arrangements, and the alignment of TACTs with current distribution and deployment practices. They also establish the need for further clinical, health economic and feasibility studies to inform decision makers about the broader implications of a transition to TACTs. They also highlight the recent emergence of ACT resistance in some African Countries and reduced efficacy, as important determinants of the market readiness for TACTs.

The manuscript is concisely written, well structured and provides adequate and important content to address the subject.

I suggest that the authors should include the questionnaire used for the interviews as supplementary data.

6. PLOS authors have the option to publish the peer review history of their article (what does this mean?). If published, this will include your full peer review and any attached files.

Reviewer #1: No

Reviewer #2: No

---

## [Author Response · Author response to Decision Letter 0]

26 Jul 2021

Note to the editor

Dear editor, 

Thank you for the opportunity to revise the manuscript. Please find enclosed a revised version of our manuscript. We would like to thank the reviewers for the detailed, constructive and useful feedback on the manuscript. Important and valid points were raised that enabled us to further strengthen the manuscript. 

Five Journal requirements were raised by the editor. We have listed them below and included our responses in italic.

1. We were asked to ensure that our manuscript meets PLOS ONE's style requirements

- We have checked the style requirements to ensure that our manuscript does meet the PLOS ONE standards.

2. We were asked to review our reference list to ensure that it is complete and correct.

- We have reviewed the reference list in order to ensure that it is complete and correct.

3. We were asked to provide additional details regarding participant consent in the ethics approvals statement

- We have now included the following statement in section 2.6 Ethical approvals and in the submission form: “Written participant consent was obtained prior to each interview and FGD. Respondents were informed about the objectives of the study and they were asked to sign a consent form. Permission to mention the affiliation of the respondent and to audio record the conversation was asked verbally.”

4. We were asked to include a copy of the interview guide used in the study as supplementary material

- In the new submission we have included the interview guides as supplementary material, both in English and in French

5. We were asked to provide additional explanation in the cover letter on why data from this study is only available upon request

- This is correct, data from this study are available upon request. Most interviews that were conducted are directly traceable to individual identities and so we cannot share the interview data without releasing the identities of our respondents. In each interview, respondents introduced themselves and they talked about their direct (working) environment. Moreover, the topics that were discussed and responses that were to our questions, could be directly linked to their job positions and affiliation, especially with higher level policy and regulatory officials. We did guarantee full anonymity to our respondents prior to data collection and so sharing the dataset would be unethical. Upon reasonable request, a list of condensed meaning units or codes can be made available by the MORU Data Access Committee (https://www.tropmedres.ac/units/moru-bangkok/bioethics-engagement/data-sharing).

Below we give a detailed point-by-point response on the issues that have been raised by the reviewers.

Reviewer 1: 

Reviewer #1: The topic is timely and vital. Plasmodium resistance to antimalarial medicines is one of the key persistent challenges in the fight against malaria. Although there is no convincing evidence of plasmodium resistance to ACTs in Nigeria and Burkina Faso, it is imperative to plan and seek ways to mitigate this challenge in the future.

The study was guided by the innovative systems approach and considered multiple stakeholders.

Response: We thank the reviewer for acknowledging the relevance of the research project and for the assessment of our manuscript. Relevant and important points were raised which helped us to improve the manuscript. Below we give a detailed point-by-point response on the issues that have been raised by the reviewer.

Respondent selection: How were participants selected? Was this a purposive or convenience sampling? Any reasons why some people from pharmaceutical/drug manufacturing industry were not interviewed?

Response: Participants for the study were purposively selected. Prior to data collection, we identified the key actor groups in the innovation system of antimalarial therapies (Figure 1) and with the help of co-authoring researchers from Nigeria and Burkina Faso we approached them for interviews / FGDs. We did not include representatives from the pharmaceutical industry because our study aimed to focus on country-level dynamics rather than global policy, industry and funding dynamics. This is explained in Section 2.2: “Our study focuses on country-level dynamics, and therefore the national borders are demarcated by the dashed line. “ We do acknowledge that the pharmaceutical industry does have a crucial and important role in antimalarial drug transitions. The perspectives of representatives of the pharmaceutical industry and other global actors is topic of another study that we are now conducting and which we aim to submit later in 2021.

Data collection Although you mentioned your semi-structured interview guide enabled flexibility, one would imagine that the questions the authors asked policymakers would be hugely different from the questions for end-users. How did you deal with this? For example, did you have separate questions for each respondent group? Did you have the same interview guides for focus groups and interviews?

Respose: We thank the reviewer for addressing this issue. Indeed, the interview guides were tailored to the respondent categories. This is explained in Section 2.2 and Section 2.3 by using the following quotes: “To prepare for data collection, we conducted a thematic analysis in which we identified the major barriers and enablers of previous antimalarial drug transitions from literature review [27]. These barriers and enablers were grouped and assigned to the key actors in the innovation system. Finally, we constructed semi-structured interview guides per key actor group to facilitate data collection [28].”

And:

 “We divided the key actor groups into three overarching categories to facilitate data analysis: Policy makers, Suppliers, and End-users. The rationale for this categorization was that they combine an overview of the innovation system with knowledge on specific parts of the system. Policy makers are best positioned to reflect on regulatory issues and policy procedures. Suppliers are most knowledgeable about distribution, retail and prescription issues, while End-users have the most relevant insights on the actual utilization of new therapies.”

In the new version of the manuscript, we have now included the interview guides for each respondent group as supplementary material. 

Data analysis: How many people coded the data? Did you evaluate the Inter-rater reliability?

Response: The interview transcripts were coded independently by two researchers who analyzed all transcripts and discussed emerging themes. The emerging themes were then also discussed with the other co-authors who had been involved in data collection and therefore were positioned to reflect on the emerging themes. This thorough and interactive data analysis procedure enhanced investor triangulation of the data and reliability of the results. 

We have now added the following text in section 2.5 Data analysis to make this more explicit: “The interview transcripts were coded independently by two researchers who analyzed all transcripts and discussed emerging themes.”

Results: The participant quotations presented illustrate the themes/findings; however the supplier category is a very diverse group . For example, the village worker may perhaps have a different opinion from a public sector prescriber. It will be helpful for the reader to have the IDs reflect this. So instead of having Supplier #22, you can have public sector prescriber #... or Village worker #.... 

Response: We thank the reviewer for this suggestion. Our study indeed reflects the position of a broad range of stakeholders with unique position in the antimalarial innovation system. In order to facilitate data collection, we decided to categorize the respondents into three broad categories: Policy Makers; Suppliers; End-users. This categorization enabled us to differentiate between the major actor groups whilst remaining concise in the storylines. 

We do agree that for the quotes that we used, it is relevant to mention in some more detail who provided that quote. Therefore, in the revised version of the manuscript, we now mention the background of the respondent in addition to the ID of the respondent that was already provided. For example we have added ‘public sector wholesaler’ in the following indication:

[“….”– Supplier 6, public sector wholesaler, BF]

Reviewer 2: 

Reviewer #2: The authors have explored the extent to which the antimalarial drug markets in African countries are ready for a transition to TACTs piloting with Nigeria and Burkina Faso. This is posited in the African setting where ACTs are still effective and TACTs would not additionally benefit the individual patient but rather future patients and the larger community by delaying or preventing artemisinin and partner drug resistance. They show that market prospects of TACTs in Nigeria and Burkina Faso will depend on the demonstration of the added value over current ACTs, their inclusion in financial arrangements, and the alignment of TACTs with current distribution and deployment practices. They also establish the need for further clinical, health economic and feasibility studies to inform decision makers about the broader implications of a transition to TACTs. They also highlight the recent emergence of ACT resistance in some African Countries and reduced efficacy, as important determinants of the market readiness for TACTs.

The manuscript is concisely written, well structured and provides adequate and important content to address the subject. 

Response:We thank the reviewer for acknowledging the relevance of the research. Also thanks for the compliments on the writing and the structuring of the manuscript.

I suggest that the authors should include the questionnaire used for the interviews as supplementary data. We thank the reviewer for this suggestion and we agree that including the questionnaires would increase transparency. Therefore we have now added the five interview guides that we used as supplementary material (both in French and in English).

---

## [Editor Report · Decision Letter 1]

10 Aug 2021

To what extent are the antimalarial markets in African countries ready for a transition to Triple Artemisinin-based Combination Therapies?

PONE-D-21-13692R1

Dear Dr. de Haan,

We’re pleased to inform you that your manuscript has been judged scientifically suitable for publication and will be formally accepted for publication once it meets all outstanding technical requirements.

Kind regards,

Lucy C. Okell

Academic Editor

PLOS ONE
---

## [Editor Report · Acceptance letter]

23 Aug 2021

PONE-D-21-13692R1 

To what extent are the antimalarial markets in African countries ready for a transition to Triple Artemisinin-based Combination Therapies? 

Dear Dr. de Haan:

I'm pleased to inform you that your manuscript has been deemed suitable for publication in PLOS ONE. Congratulations! Your manuscript is now with our production department. 

Kind regards, 

on behalf of

Dr. Lucy C. Okell 

Academic Editor

PLOS ONE